# Factors Affecting University Choice Behaviour in the UK Higher Education

**Dean Heathcote [1], Simon Savage [2] and Amin Hosseinian-Far [3],***

1   Quality Department, Bradford College, Bradford BD7 1AY, UK; d.heathcote@bradfordcollege.ac.uk
2   School of Computing and Communications, Faculty of Science, Technology, Engineering and Mathematics, The Open University, Milton Keynes MK7 6AA, UK; s.a.savage@open.ac.uk
3   Department of Business Systems and Operations, Faculty of Business and Law, University of Northampton, Northampton NN1 5PH, UK
*   Correspondence: amin.hosseinian-far@northampton.ac.uk; Tel.: +44-016-048-93-628

**Abstract:** Although regulations and established practices in academia have focused on a data-rich model of performance information, both to evidence operational capability and to support recruitment, it is considered that this approach has been largely ineffective in addressing student choice behaviour. Historical studies, business, psychology, and technology theories have pointed to the oversimplification that a data-led strategy can result in for mapping human behaviour. In this case, decision processes of students are understood to be nuanced by a vast range of factors with variable relevance for everyone. The effort of the study is to address how Systems Thinking, related policy development, and associated enabling techniques can be applied to the field to provide both a deeper understanding of the dictates of student behaviour and, by extension, the appropriate foci for data provision, enabling comparative business performance assessment of Higher Education Providers. This research has followed the Design Science Research (DSR) methodology; the developed model has been successfully evaluated against the understanding of education practitioners in an interview consultation process of the methodology. The analysis of interview feedback and the development and refinement of the proposed model generate the principle findings of the study. The model outlines the factors that might affect students' choices in the UK Higher Education.

**Keywords:** higher education policy; systems thinking; design science research; performance data

## 1. Introduction

There is a recognised problem with the presentation of performance data by academic institutions, partly caused by the regulatory preconceptions that have attempted to dictate what should be presented to students [1]. In the UK, provision of information to prospective students has been mired by poor interpretation of the requirements of the consumers it is attempting to serve, and it is only in the last few years that serious efforts have been made to understand and address the needs of the client group. For too long, both educational overwatch and the institutes themselves have presumed to understand the informational needs of the market, whilst recent research has shown that it has tended to saturate the consumer with the wrong data, or made those data too complex or obscure to be useful [2].

This has been endemic in two government-led initiatives—and the ways in which they have been utilised by Higher Education Providers (HEIs)—Unistats and the National Student Survey (NSS). Each is intended to provide comparable performance feedback to institutes and to help inform student choice in the recruitment/selection process.

This piece of research attempts to identify and structure the factors that are affecting student choice in the UK higher education system. It also offers methodological novelty by adopting the

Design Science Research (DSR) methodology to visualise a causal model embracing all the identified factors. This paper is organised as follows: Section 2 provides the research background and elaborates on the problem domain. Section 3 similarly provides a review of key literature with a view to identify the variables for the initial model. Sections 4 and 5 outline the DSR methodology and, where required, reflect on some of the key overall responses as part of the interview data collection method. Results including the final models and a discussion on the models are provided in Section 6. The paper is book-ended by a conclusion section in Section 7.

## 2. Background and the Problem Domain

### 2.1. A Regulatory Conception of a Buyer's Market

A dichotomy exists in academic recruitment. On the one hand, institutes have long held preconceptions as to what information is most important to prospective students and, therefore, what performance data is most relevant to consumers. On the other, there is the actual, pragmatic, esoteric, and sometimes illogical, emotional response of prospective students to the challenge of selecting academic study or an institute at which to study—a set of decisions that often have little in common with the rational choice espoused by theoreticians [2].

Given the stately pace of government initiatives, relatively recent strategic material, such as the government White Papers on Higher Education by the Department for Education and Skills, UK and the Department for Business, Innovation, and Skills [3,4], operate on a presumption of a purely rational decision-making process that clearly equates quality delivery to reward and implicitly rational student choice. The 2011 White Paper cites [5], in particular, for a list of quality-defining traits, each of which is clearly underpinned by quantifiable dynamics: Class and cohort size, contact with academics, student effort and engagement, feedback and, teacher experience and quality. The resultant foundations served to justify the development of a Key Information Set (KIS) of government-approved and regulated data that institutes were required to record and share with students, current and prospective, of course being made publicly available as a result.

In March 2017, the Higher Education Funding Council for England (HEFCE) publications of guidance on information for applicants [6] focused on course structure and module information presented within prospectuses or online. Low priority is given to environmental factors such as location—a duty to inform students of physical distances between campus locations and accommodation, for example—but, for the most part, the concern falls on the syllabus. It is admirable that the directive encourages a more uniform and comprehensive presentation of these data, but it does sacrifice wider contextual factors and may provide more material than any student will utilise prior to a decision having been made. This decision is attributed to the review of Unistats and the KIS, and to the CFE Research Review of Information about Higher Education (specifically [7]).

Despite that the regulations and established practices in academia have focused on a data-rich model of performance information to evidence operational capability and to support recruitment, this data-led approach is widely argued to be ineffective in addressing students' choice behaviours [8,9]. Therefore, this piece of research seeks to answer the following research question: What factors interplay with the student decision-making process? Students are, in this treatment, seen as the heart of the decision-making cypher, and so it is vital to survey the problem of choice from their standpoint. Ultimately, any actions by external bodies can only influence their action, never dictate it. By this logic, for the purposes of this study, the prospective student stands as the primary 'Customer' within the system. However, this is not to say that institutes cannot be viewed as the beneficiaries of the model as a whole in their own way, too.

### 2.2. Critique of the Rational Model—Regulators

Policy and regulation as well as institutional ideology have, to date, focused on the student as a purely rational entity, and thus largely delivered information to support rational decision-making.

However, this increasingly is shown to fall short of the realities of student behaviour, and the standard pattern of data made available to students is not seen as fit for purpose. Despite the wide variety of data made available—increasingly online, but via traditional mediums also—critics have noted that students are not necessarily using the available data at all [10].

Howson and Buckley provide a history of quantitative measures from the students' success [11], yet the added value of such measures for various stakeholders is unclear. Performance data as presented, rankings, and reviews all serve greater importance to institutes than to applicants. Moreover, HEFCE has established that rankings are of more value to the institutes themselves [12]. Numerous sources critiquing the use of rankings as benchmarks of HEI performance, from the late 2000's onwards, nevertheless fell on stony ground with most regulators at the time, whilst there remains a place for such measures in regulations; their influence on student behaviour in application to institutes remained overstated. Such data, however illuminating, had little proven impact on student application behaviour [12]. Given the 'strong persistence' property of student behaviour, whilst the impact of ranking on teaching and institutional position is substantial, it has a minimal impact on student behaviour.

Similar issues can be raised with National Student Survey (NSS) data; first introduced in 2005, over that time, the NSS has remained largely a stable set of satisfaction data, relating the experience of final-year undergraduate students in UK HEIs. Furthermore, the effect of the survey results on student applicant behaviour (a UK-specific piece, but relating to international work due to the paucity of similar activity to date) is shown by [13].

There is a dissonance between what institutes and regulatory bodies have considered important to prospective students and what appears to matter. For law students, at least, university choice is influenced by notions of prestige and reputation [14], which have little if anything to do with actual performance or user satisfaction, to which any change in performance carries a notable lag before having a smaller impact upon rankings and reputations [13].

'Students at the Heart of the System', introduced by the UK Department for Business, Innovation, and Skills [4], introduced numerous reforms to the Higher Education (HE) system; however, it remained wedded to the concepts of the student as a rational decision-maker, and consequently of data as driving the decision. However, with Higher Education being seen as a market economy, notions such as students' voices can be deciphered differently [15]. Despite commitments to make the market more flexible and to make entry more straightforward in other respects, there did not appear to be a recognition of the decision process of the students themselves as being inherently complex, rather considering the priority to be 'high-quality' information [4].

To justify this rationale, the report cites institutional research, including collaborative and independent work. Moreover, 51 information items (factors) which are considered important in Higher Education student choice are listed by [16]; these were led by satisfaction with teaching standards and courses. However, these were only 'useful' to just over 50% of respondents (54.4% and 50.5%, respectively); the report naturally favours those with higher results, but does not focus on 'lesser factors', and it is interesting to consider which are, or are not, deemed valuable to the KIS dataset.

Nevertheless, warnings are still being made by critics of this model, sounding out against an over-reliance on data for data's sake. In an open letter to the new head of the Office for Students, Sir Michael Barber, appeals against a simple reliance on the Metrics and Deliverables apparently being adopted and reaffirmed by the new commission are being made. A number of factors are also noted by [17], including an inclusive vision, public engagement, investment, and corporate educational responsibility, which are as vital to quality delivery and regulation as market competition and standardised comparison [17]. That this point has to be made in the face of the upcoming regulatory body that is presumably intended to direct the future of education is only an indication of the refusal of the regulators to take on board criticism and evidence that the current model is not wholly fit for purpose.

## 3. Variables and Relative Importance

A number of key works have investigated the problem explicitly from a student perspective. The impact of rankings on applicant behaviour was examined by [18]. Whilst a 10-place fall in rankings affects applicant behaviour only slightly overall—declines of 0.5%–0.9%—the impact for high-prestige institutes was far more marked, with the same modification eliciting a 2.7% to 6.2% fall in applications, depending on the source [18]. Moreover, ranking changes impact tariff scores of students, suggesting that lower institutional performance has a small, but definite, impact upon the quality of applicants approaching the institution. Elements such as these are, of course, ripe for modelling in a dynamic environment.

Awareness of Unistats was surveyed by [7], highlighting by extension the primary sources and, therefore, decision-driving factors [7]. Only 42.5% of prospective students had heard of Unistats, and of these, over 50% encountered it via either the Universities and Colleges Admissions Service (UCAS) [19] or an online search, and less than half had been advised of it by teachers or found it via institute sites themselves. Only 14.4% of the students indicated that Unistats was their most important source of material, though there is a strong indication that many found the site too late for it to influence their decision. 'Visibility' of the Unistats site is amongst the criticisms levelled elsewhere [20].

Motivation of choice is arguably the overlooked factor; students have a framework of support, but also a mental landscape in which rational, data-driven options are only one element among many. Research approaches that treat the market as a single homogeneous entity are ultimately flawed [21], and, in particular, divide the various motivating elements into the two principal strata of 'student factors' and 'institutional and student–instructional factors', as can be seen in Figures 1 and 2, respectively.

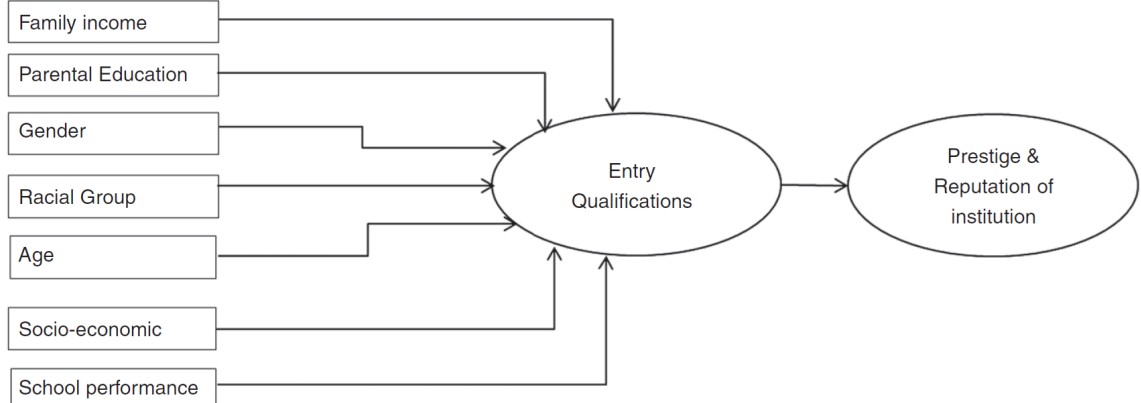

**Figure 1.** Student characteristics [21].

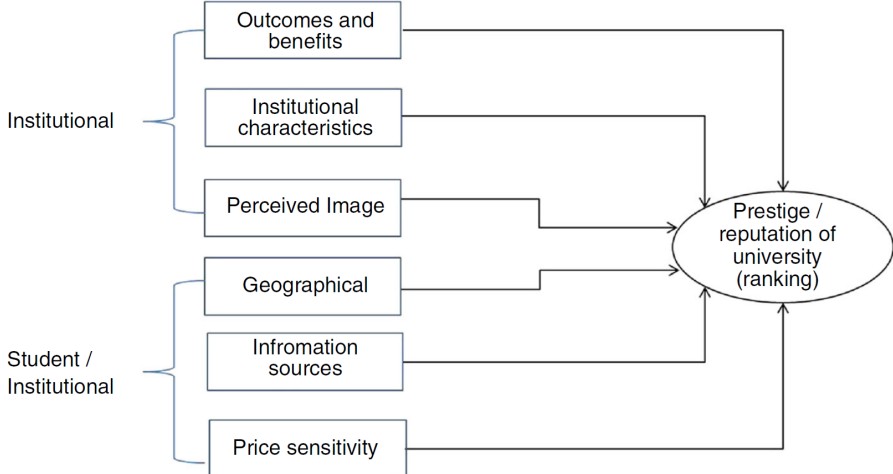

**Figure 2.** Student–institutional and institutional characteristics [21].

As can be seen, the implication here is that many factors outside of traditional performance data influence the perceived prestige of any institute; indeed, the authors additionally list 42 elements as influence factors. These include such critical elements as course content, fees, and facilities, as well as esoteric personal elements, such as proximity to home, peer influence, response speed to applications, and the visual appearance of a campus.

Reflecting upon experience, 10% of students stated that they would definitely change courses based on their experience at their university/course of choice [22], with a further 24% in a survey set of over 15,000 indicating that it was a real possibility. The work of the Higher Education Policy Institute (HEPI) highlights another element: That the decisions students initially make, and the rationalisation that justifies them, can often prove to lead to the wrong choices. The material to inform is not appropriate and may influence that erroneous decision, alongside the belief models of the students themselves.

These belief models were most clearly reflected when students were asked on what institutes should save money—with sporting and social facilities and buildings ranked at the top with 46% and 45%, respectively. Conversely, less than 10% wanted to see academic development of staff, teaching hours, or learning facilities reduced, at least reinforcing the students' commitment to the quality of (their own) teaching delivery.

## 4. Methodology

Development of the causal artefact was accomplished using the Design Science Research (DSR) methodology (Figure 3). The methodology was initially developed as an approach in the field of information systems; nonetheless, building upon existing design knowledge, use of DSR found its way into other disciplines and applications (e.g., [23,24]). There are a few varieties of DSR; Montasari [24] argues that the one introduced by Peffer [25] (also adopted in our research) is the most comprehensive. The artefact development was conducted using a systemic approach called Causal Loop Diagramming (CLD). This causal modelling technique provides the opportunity to illustrate variables (factors) and depict causal links (either positive or negative) that denote the balancing or re-enforcing nature of each causality. The causal model variables were extracted through narratives, as outlined in the next section. Considering the methodology, the interview iterations began to optimise the developed artefact. Details are outlined within the interview consultation section of this paper. Three rounds of interviews, i.e., a total of 15 interviews, resulted in the complete proposed models, with no further comments to add.

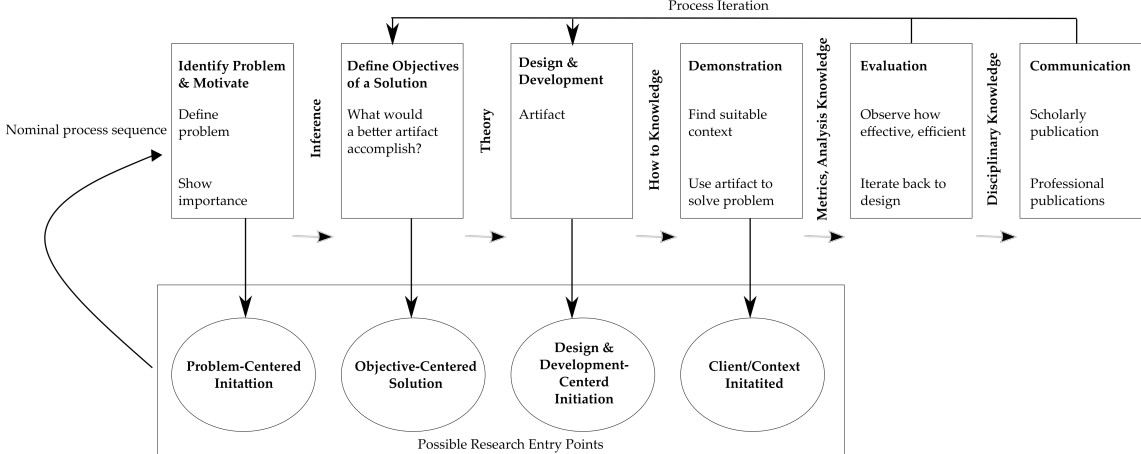

**Figure 3.** Design Science Research methodology, adapted from [25].

The modelling technique adopted for developing the artefact in this context is Causal Loop Diagramming (CLD), which is a modelling element of 'System Dynamics' as a discipline. Modern Systems Thinking is advanced by a number of authors and organisations, and has become widely popular in the business and technology fields, developing into the 'System Dynamics' approach of the last twenty years, Kirkwood and Sterman being advocates of this wider approach ([26,27]). The Integrated Development Environment (IDE) used to construct the CLD was Vensim PLE version 7.2.

Causal Loop Diagrams (CLD) and feedback simulation can traced back to the early 1970s, though with no certain genesis [26]. CLD is a particularly useful tool in the IT sector, and is generally implemented via computer modelling, as it will be in this study. It is possible to replicate in a manual model or through experiments, but the complexity of change is easier to accommodate digitally. What CLDs concentrate upon is the relationship between positive and negative factors, and between 'reinforcing', positive loops and 'balancing', negative loops. The combination of these in a systems context generates a predictive model, commonly used for 'Stock Prediction', but these are not limited to economic contexts, and can equally be applied to social, environmental, and other scenarios.

*4.1. Vensim—A Methodological Tool*

Developed by Ventana Systems, Vensim is the selected tool for developing a CLD framework for review. Whilst containing all of the key tools for model building and Systems Thinking methodologies, it also benefits from being freely available in a full-featured academic version. The specific version employed is PLE 6.4E (x32). It is a graphical modelling program, which can be driven by imported data or allow for the manual creation of a model, as is the case here, where irregularly represented source data have been drawn from various research papers and regulatory documents.

Whilst featuring strong analytical tools that will not all fall into the remit of this study, Vensim can, e.g., handle Stock and Flow predictive models, and offers algorithmic analyses, such as Linear Quadratic Estimation and Markov Chain probability distribution (MCMC). Vensim can also export simple visualisations of data for those without access to the software for peer review. To this end, it is particularly useful in the engagement of groups without prior experience of CLDs who may, in turn, have limited familiarity with Systems Thinking, but a grasp of the fundamental logic of operational issues.

*4.2. The Consultation Process*

The narrative extraction approach taken was to develop an initial model based on the CATWOE (Customers, Actors, Transformation, Weltanschauung, Owner, and Environment) analysis of primary research. CATWOE is a fundamental analysis technique within the Soft Systems Methodology approach. As a problem evaluation tool, it seeks to represent all of the factors influencing a scenario [28]. This built a test diagram within Vensim, seeking to make an initial attempt to encompass all of the identified variables, and to project their initial structural form and relative importance to the design and, consequently, to the decision process.

This proposed design, along with tabular representations of the factors—attempting to reflect the relative importance of the positions of variables within major factorial groups—was then provided to the consultative members in advance of the interview process to provide them with an opportunity to examine the design. Whilst provided in a Vensim model format as a matter of course, a clone of the diagram was also provided in an accessible format (i.e., MS PowerPoint, via image shots with supporting dialogue) for those without access to or familiarity with the software.

Consultation on the model was conducted as part of a three-stage interview process, with the second part of the dialogue concentrating on the model itself. This was preceded by discussion of the interviewees' conception of the operational market, and followed with discussion of information use in the academic recruitment market, as well as some investigation of how they feel specific institutional decisions may impact student interest.

Fundamentally, the consultation over the proposed framework with numerous stakeholders within the HE sector will fulfil the 'demonstration' and 'evaluation' stages of the DSR methodology. Once initial consultation is completed, the design will pass through a second iteration, returned to the group for further consideration, now informed by the discourse at the first stage, as well as incorporating their recommendations and thoughts on the design. Additional questioning will be developed at this stage to address the findings of the primary interview and the evolving model. This will result in the final framework.

*4.3. The Initial Model*

Through narrative extraction, innumerable features were identified. In total, an initial model identified 82 distinct elements of influence across nine major themes.

1.　Course Suitability—content, entry tariff, delivery style, etc.
2.　Locality—distance from home, appeal of destination city, transit network [29], etc.
3.　Fear of Debt—fees, living costs, prospects, etc.
4.　Financial Relief—scholarships, bursaries, benefit support, etc.
5.　Current Reputation—historic reputation of institute, opinion of peers, image, national rankings, etc.
6.　Pragmatics—classification of institute, how work is expected to be handled, ability of institute to respond well to enquiries, etc.
7.　Flexibility of Programme—part-time options, sandwich study, attendance hours, etc.
8.　Appeal of Institute—resources, class size, teacher qualifications, provision of lifestyle enhancements, etc.
9.　Demographics—applicant age, gender, ethnicity, religion, etc. These play an often ill-regarded element in many fields.

Within these, nine key iterative loops were identified, though not necessarily on a like-for-like basis.

1.　The Course Content—Module Options. Positive (reinforcing) cycle.
2.　The Proximity of Contact—Distance from Home cycles. Two interrelated balancing cycles.
3.　The Accommodation Cost—Quality of Accommodation. Positive cycle.
4.　The Employment Opportunities—Employment Rate—Earnings. Positive cycle

5.   The Family Income—Socio-Economic Status. Positive cycle. The factors are influenced, in turn, by the employment cycle, carrying considerable system delays in impact.
6.   The Historic Reputation—Ranking of Institute. Positive cycle; again, reflecting a delay mechanism for interrelated influence.
7.   Library/Resource Facilities—Areas for Quiet Study—Library/IT Quality. Positive cycle.
8.   Attendance Days—Active Hours. Balancing cycle; a poor organisation of academic time will reduce enthusiasm for a course.

This first design displayed the original sources for the data in a colour-coded format (Table 1), with a number of additional rules relating to the use of underscored, bold, and enlarged fonts to indicate relative importance. These reflected a mix of authorial sources:

**Table 1.** Colour-Coding approach.

| Color | Source |
| --- | --- |
| Black | Bowes et al. (2015): Main text [7] |
| Blue | Price, Smith, and Agahi (2003) [30] |
| Red | Hemsley-Brown and Oplatka (2015) [21] |
| Purple | Buckley et al. (2015) [22] |
| Light Blue | Bowes et al. (2015): CFE Research [7] |
| Orange | Velouttsou and Paton (2005) [31] |

These were augmented by the authors' own findings based on their sector experience. It is no coincidence that many of the sources are very recent; research on integrating human behaviour with the dynamics of data provision and the business-derived concepts of performance management has only recently begun to surface. This model would go on to be a major part of and basis for the consultation with the interview group of consulting experts in the field.

*4.4. Interview Consultation*

Five key interviews with individuals involved in the sector were conducted in three rounds with a view to optimise the model. Each interview comprised the same core set of questions (Appendix A), with a flexible range of additional questions and comments intended to elicit further responses, which was added as was deemed fit by the interviewer within the context of the session. This resulted in interviews of typically 45–60 min in length.

On either side of the discussion of the CLD model, interviewees were probed further on the matter of context. There were many concerns about the lack of contextualisation in regulatory data, where institutes could not present a balanced perspective. As data are only presented in a common, purely analytical form, there is little to no allowance in either the KIS data or by utilising models—such as Unistats and the new Teaching Excellence Framework (TEF)—for distinction between institutional attributes in terms of attainment, financial support, or specialism. Interviewees felt that students are left to inform themselves of context from other sources, and whilst it may be easy to make broad presumptions of the difference between a 'Russell Group' university, a college-based provider, a dedicated music or dance conservatoire, or other providers, when there is less distinction implicit in the title or reputation, a prospective student is ill-served by the numbers alone.

Of course, there is a recognition that contextualisation and elective choice of published data would lead to manipulation of material, but it was also felt by many respondents that this may be needed to counter the same negative interpretations from press and other media.

Furthermore, there was an identification of the gaps in data provision as it stands; there was no coverage of vocational placements, for example, but instead a focus on traditional research activity, which is not relevant to some lower-tier institutes. Fundamentally, the bias within the system towards

traditional institutes and traditional course delivery was seen as in dire need of addressing, but it was recognised that to do so could open up a separate set of biases.

### 4.5. Interviewees' Profiles

The five interviewees all worked within education, in a variety of roles, and represented a range of institutes:

- A Course Leader and Teaching Fellow for a regional metropolitan university in northern England;
- A Principal Lecturer, coordinating the recruitment activities of their faculty at a regional metropolitan university in northern England;
- The Head of Marketing and Recruitment for Further Education (FE) and Higher Education (HE) provisions at a northern regional college in England;
- The Director of Higher Education Participation and Skills for a metropolitan college-based HE provider in the northeast England; recently of a role with another College-Based Higher Education (CBHE) provider in one of the country's largest cities;
- The Market Research Manager for a regional metropolitan university in northern England.

This permitted a range of perspectives to be drawn on the content of the CLD and the working context it attempted to represent. It is notable that the group, being of a small size, does not contain a full diversity of institute types, comprising post-1992 institutes and college-based higher education providers. There are no older 'Plate Glass' or 'Redbrick' providers in the set, and certainly no identifiably premium institutes (Redbrick and Premium institutes are institutes from approximately the 1950s to 1990s, and from the late 19th to mid-20th centuries, respectively). One concern for the study was the difficulty in engaging institutes themselves in a relatively short timeframe. A total 63 staff members across 43 institutes were approached, with only 5% engagement rate.

### 4.6. Institutes' Views of their Own Provision and the Wider Market

Interviewees were asked how they felt their institutes appealed to students, and were invited to comment on whether they felt this varied from the national norm. Naturally, all respondents considered the course itself as a key factor, but beyond this, a number of other initial thoughts were prevalent. For the metropolitan university, respondents' reputations and locations were seen as important draws, even though the former was recognised as a challenge compared to higher-profile institutes, and the latter was governed largely by the success and popularity of the city they were located in, to which the institute contributed proportionately little. Reputation was seen as tied to offers in one case, whilst making a proactive choice about living at or away from home was alluded to by all; the view appeared to be that metropolitan universities draw from a more localised audience than major national institutes. The respondents with teaching experience pointed to the quality of facilities as a motivating factor, whilst both teaching and marketing would ultimately reference the importance of strong communication in the pre-/post-application process.

All three felt that the experiences of their institutes were typical of national trends. One in particular noted that, in their opinion, it was difficult to establish why any student rejected options post hoc, observing that when asked for their reasons, a student was as likely to hide their true reasons as to reveal any insight. Conversely, the marketing representative pointed to evidence from decliners, indicating the it is a fine balance of many factors influencing decisions, although demography plays a part in these choices; each personal case is unique, and they felt this to be consistent with national trends.

The other interviewees, representing CBHE providers, felt that their client bases had some strongly divergent demands. Many of their students were internal conversions from FE provision, and location was essentially not a choice. Courses remained important, but that choice was more likely to be vocational and, of course, a natural progression from the further education provision of the institute. Both noted a sense of security and familiarity—'comfort'—as being very important to their students,

one stating that university attendance was a negative, intimidating proposition for them. Where external students are attracted, it is as a result of course specialisms and investment in new—and high-profile—facilities.

The national perspective is tempered by the distinct fee and structural differences from major HE providers, and here, naturally, the CBHE members recognise more distinctions. However, financial concerns were repeatedly downplayed as a motivating factor by all parties, and the CBHE interviewees reinforced the point that students coming to them were more likely to be local and more comfortable with a college context. One in particular noted that the FE college context would not appeal to many students looking for a 'university experience', but that for students from low-participation backgrounds, their more vocational priorities were more appealing.

*4.7. In Response to the Causal Loop Diagram*

The interviewees the diagram comprehensible on the whole found, but it was the case that all used representative versions of the diagram developed in PowerPoint for easy consultation, rather than using the Vensim model itself. None of the representatives had any prior experience with Vensim, and only one of them showed any real familiarity with the CLD concept beforehand; that said, this did not prove to be a handicap in terms of comprehension or discourse. From the standpoint of the study objectives, this raises a question of implementation, which will be returned to in the subsequent evaluation.

Four out of five of the interviewees found the initial diagram quite comprehensive, but not without some issues, the most obvious of which was the sheer size of the initial diagram; opinions included needing a long time to digest all the content, initially appearing daunting, and wondering whether all students would make use of every variable. Of course, this latter point is not the expectation of the design; rather, it attempts to incorporate every possible variable that may influence thinking.

Nevertheless, whilst comprehensive, there were a number of elements that the interviewees felt were underplayed or absent from the model in its first iteration. These included a mix of social and structural factors. One interviewee considered the ability for students to make friends as a separate element from established connections and not properly reflected in the established appeal of institute or city. Tied to this, they felt that institute-led initiatives to organise study-relevant activities for students would be important as a social networking tool.

Three interviewees touched on the importance of open days and institute visits by students, and in hand with this, the crucial influence, in their view, of their student ambassadors. Carefully selected representatives of the course were seen as a particularly effective way of conveying both the course strengths and the reality of the institute experience to prospective applicants in a way that other means would struggle to achieve. Direct exposure to the campus and the staff has the potential to bond a student to the institute from an early stage. However, this went hand in hand with a recognised, if somewhat amorphous, risk of negative incidental experience. Relatively intangible and, in many respects, idiosyncratic factors could result in a candidate coming away more deterred than enthused. Examples suggesting fear for personal safety, disappointment with maintenance, or levels of personal contact were cited.

The difficulty with this is, of course, how to incorporate it into the model, as negative subjective experience could effectively impact any factor; however, the concerns of the interviewees leaned towards the personal exposure to the institute as a priority, indicating a believe that the other elements could largely be addressed by other efforts (such as quality and presentation of online information) or were entirely outside of realms an institute could influence, at least in the short term.

Recognition that elements of import to older and younger students were distinct was common, which points again to certain core beliefs in the market—notably, that financial concerns are more or less irrelevant to younger students. Despite the open rebellion at the prospect of increased tuition fees in the 2010–2011 period [32], it quickly became accepted as the status quo, and appears to have had little long-term impact on student numbers [33]. There is a general consensus amongst the interviewees and

beyond that younger students are aware enough to understand that the debt model of undergraduate study is one with more benefit than loss, whilst such concerns are more prevalent amongst mature students that are tied to an area or looking to fund postgraduate study. Economic concerns, as more pragmatic variables, are consequently the precinct of generally older students, and those with less social precedence for higher education.

Although peer influence is consistently mentioned in the model, at least one interviewee specifically highlighted the influence of employers and, by extension, other sponsors on student choice. Ultimately, any study activity funded by an employer is likely to be determined by the demands of that party, far less so by the students' own tastes, though connection to career progression must be presumed as a shared goal at least.

Finally, course- and reputation-supported specialisms were mentioned in several cases. It is recognised that the specialism of an institute can undercut the general perception of a provider in other respects. Many institutes—such as specialist arts and business colleges—can operate essentially outside of normal league table concerns due to their focused provision, whilst other institutes may have schools that are held in far higher regard than the rest of their provision. For such institutes or faculties, the normal rules and, indeed, the bulk of the model, may not apply, as it is likely that a student's choice is driven by specific provisions meeting their aims more than any other element. Of course, this is likely to mean that these are students for whom some of the other variables are wholly redundant, and implies that students have the freedom to be mobile and highly flexible in their study choices.

The overall structure and links within the design were generally well received, with only some minor suggestions regarding the adding of further links. Such a response indicated, however, that for some, the model was so complex that they possibly did not have time to unpack all the details.

## 5. Scenarios and Future Pressures

To draw opinions on how specific institutional actions/outcomes outside of data provision were perceived as likely to influence students, three scenarios were briefly presented to the interviewees. These, relating to reductions in entry tariffs, fees, and institutional performance in league tables, were intended to gauge the priority ascribed to these variables within the model by the interviewees.

The status of the participating institutes at the mid-lower end of the tariff and reputation scale was apparent in the response, and so the answers here could not be taken without challenge as indicative of the entire market; nevertheless, they displayed a considerable consensus and strongly indicated the sentiments of the sector they reflected.

In terms of lowering tariffs of entry, the university respondents felt that this would increase the number of applications to their institute; however, this came with a concern that it may impact the quality of applicant the institute would receive, and that—in the longer term—this may impact the reputations of institutes. One of the interviewees admitted that this was an experiment their faculty was to conduct in the next year, motivated by the changing market and external competitive force since the removal of the HEFCE cap system limiting institutional recruitment.

For College-Based Higher Education (CBHE) institutes, who already had much lower tariffs, lowering these further was seen as having little effect, effectively of no credible value. Reversing the situation, a rise in entry criteria was perceived as likely to exclude part of their key demographic—low-participation groups. Again, entry tariffs were seen as strongly linked to institutional repute, with higher criteria recognised as an indicator of quality; major changes—especially increases—in tariffs were seen as practicable only as part of a supported rebranding exercise. Additionally, for the Further Education Colleges (FECs), a concern would be that increased tariffs would bring them into direct competition with universities for students, where, again, profile and reputation would limit their success.

One interviewee also spoke about the relative value of unconditional offers, where high-tariff/high-profile institutes could use these to lock in desirable applicants, whilst lower-profile

institutes could not expect the same draw. This raised the side issue of the 'false' conditional offer, a 'contract' all too readily waived if a student does not achieve the required grades, in favour of making a generous fall-back offer.

In terms of fees, a postulated reduction to the value of £1500 was proposed—sitting squarely between the present maximum fee for institutes with access agreements and those without. Currently, it is the industry norm for English universities to charge UK nationals the maximum (or very close to it) in tuition charges. The government's expectation prior to introduction had been that only the top-tier 'Russell Group' institutes would be able to justify such high costs; however, the market rapidly moved to the maximum value as standard, many mid- to high-ranking institutes feeling justified in charging the maximum fees immediately, and others following suit as price became a barometer of quality in its own right. A total of 74% of all institutes and 98% of universities charged the maximum fees in 2014, just two years after the introduction of the new rates [34], and as previously noted, application rates were recovering to near record levels, whilst acceptances had already reached record levels [35]; financial concerns were not, at least, stopping students entering education.

However, from the standpoint of the interview group, fees are an inexact science. For all, it was uncertain how much effect a reduction could have. As noted in one case, a 10% price difference was seen as inconsequential for an audience who largely did not perceive the reality of the debt; rather, the main concern was moving out of line with one's peers. In this respect, whilst lower fees could be seen as appealing more to financially aware students—mature and less affluent individuals—the reaction from others may be to question the quality of provision if it is considerably cheaper than the industry standard. Differentiation was alluded to from the opposite perspective for CBHE providers, in that to justify charging above the £6000 level, they have to have considerable reputation, facilities, or course specialisms capable of justifying the investment on a student's part. In short, it is not the case that students worry about the cost of a course, but they are perceived as sensitive to its value.

The final scenario—regarding a notional fall in university rankings –proved to need immediate modification to garner a useful response, as each institute already considered itself in the lower quartiles of league tables and performance rankings. Though four out of five interviewees had already expressed little regard for such rankings and had questioned how much attention students paid to such material, in general, they felt that an improvement in their league table positions—even if relevant to only a single survey—may have a marked beneficial impact on their queries.

As noted by one respondent, the arbitrary structure of quartile rankings could easily see all institutes meet performance benchmarks, and all may achieve excellent levels of retention or student achievement; when the data are forced into divisions, this creates meaningless distinctions that ill-serve the applicant. Another added that, as the contexts to negative rankings are not provided, students may not be able to determine the significance of a problem; current students may not recognise a problem with the course. Such rankings were certainly seen to have little influence on lower-achieving and otherwise constrained students, typical of large portions of the interviewees' intake, for whom choice of institute is more limited.

Finally, interviewees were asked what they envisaged for the short- to mid-term future in terms of pressures on the recruitment system. Most of the group mentioned demographic pressures as a major concern; citing the population changes up to 2020, they expected to see the national proportion of 16–18 year olds fall consistently, in partnership with an envisaged fall in international students as a result of the exit from the European Union and changing visa practices. This contributed to a sense of a constricting market for the institutes involved. Additionally noted was the general transformation of learners into consumers and their consequent increased expectation of personal provision. Financially, this placed advantages with the larger, high-reputation institutes to offer more tailored provision at a price seen as assuring quality.

Conversely, Higher-Level Apprenticeships (HLAs) were discussed as a future opportunity for low- to mid-tier institutes. These offer vocational routes in higher education and can lead to a degree qualification by a longer track. As yet, they are not operational, but at least one of the institutes in

the survey was in the process of presenting proposed courses. A concern, however, was that the upper-tier, traditional institutes would also see the benefit of offering these courses, and the available capacity would soon find itself being absorbed by groups for which it was never intended. As HLAs are ultimately a partnership activity with the private sector, businesses may come to determine that only the most able students are welcome, rather than the students with less academic aptitude that the qualifications were intended to target.

## 6. Iterative Models: Refinement and Revision

Reviewing the material from the interviews, it was apparent that the initial model was strong and reflected many, though not all, of the perceived influences on the system well. This in itself provided a strong indication that sector experience chimed with the findings of established research, with both taking an often divergent stand to the regulatory position of analytical data above all else.

In terms of the static factorial model, only a few additions were deemed necessary at this stage; however, some rearrangement of the model to allow for clear linkage of certain items was also required. Secondarily, the design was clarified in terms of the previous colour design and use of font sizes to highlight importance. This process of enrichment permits the system to identify not only where additional features are needed, but where due importance is placed within the systems and which elements may prove to be of less relevance overall [36]. The second design consequently varies more in layout than any other element. However, subtle changes, in fact, confer a major refinement of the model as a whole.

This second model was circulated to the interview group for consultation, but in parallel, the requirement of a division of the single model into three specific demographic subtypes was identified: One for young entrants (the 'standard-path' 18–20 year olds), another for mature students (postgraduates and returners to education) for whom certain issues were identified as of far greater importance, and a third to cater to the specifics of an international audience, whose choice parameters most closely align to the regulators' presumptions of student behaviour, and are resultantly widely divergent from the domestic audience.

The final single model that was developed after the second round of interviews is presented in Figure 4.

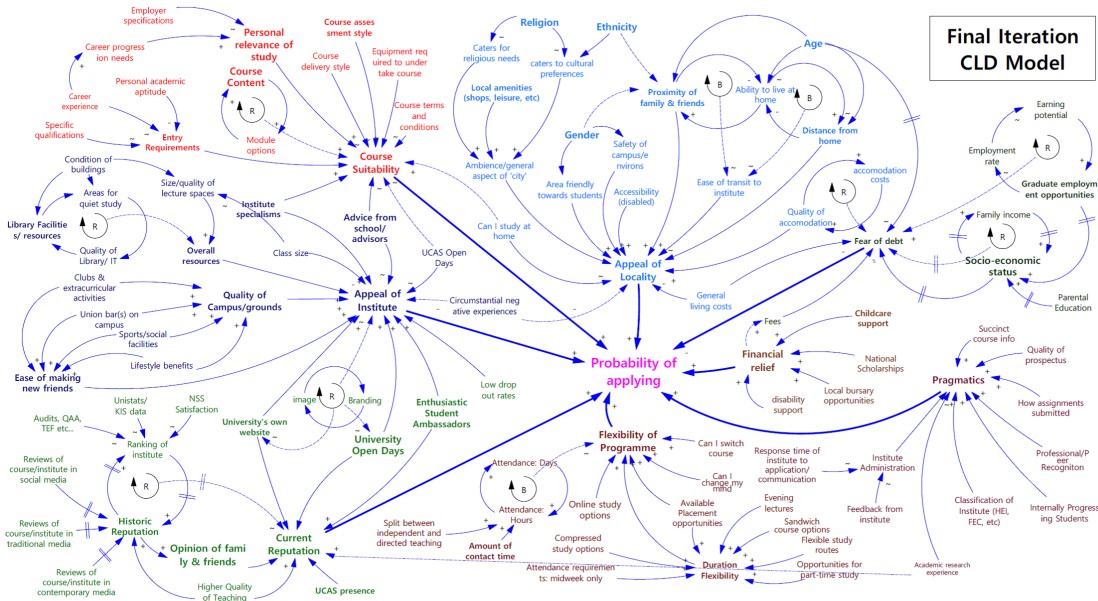

**Figure 4.** Single Causal Loop Diagram (CLD) model irrespective of students' characteristics.

As highlighted earlier, it became apparent that a single perspective of the factors influencing students' choice is unable to embrace all specific requirements of the subtype. Therefore, three additional subtype models were developed to entail the minor differences that cater to specific types of students. The causal loop diagrams considering the three groups—mature, international, and young students—are presented in Figures 5–7, respectively.

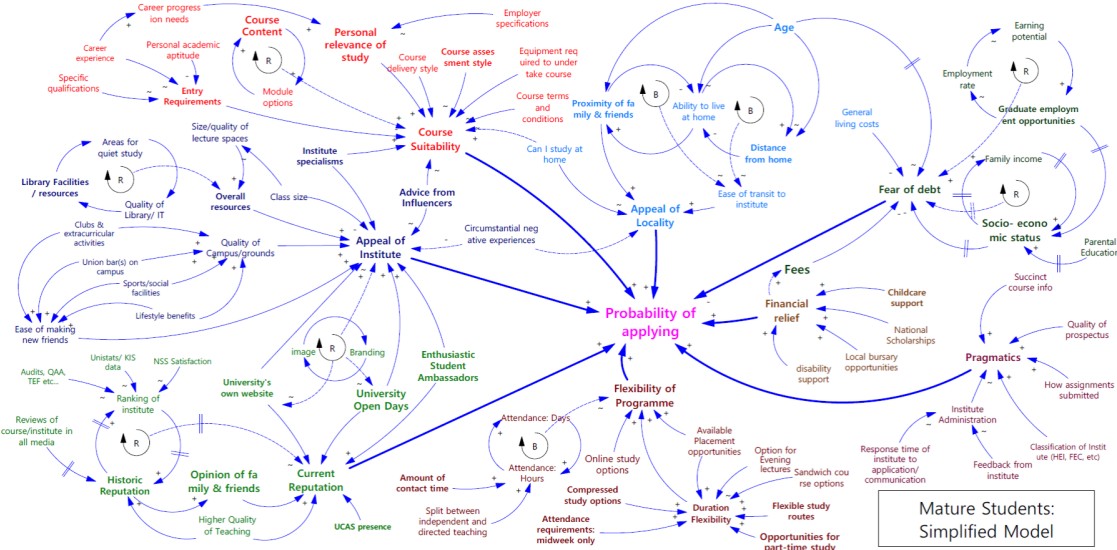

**Figure 5.** Final CLD model—mature students.

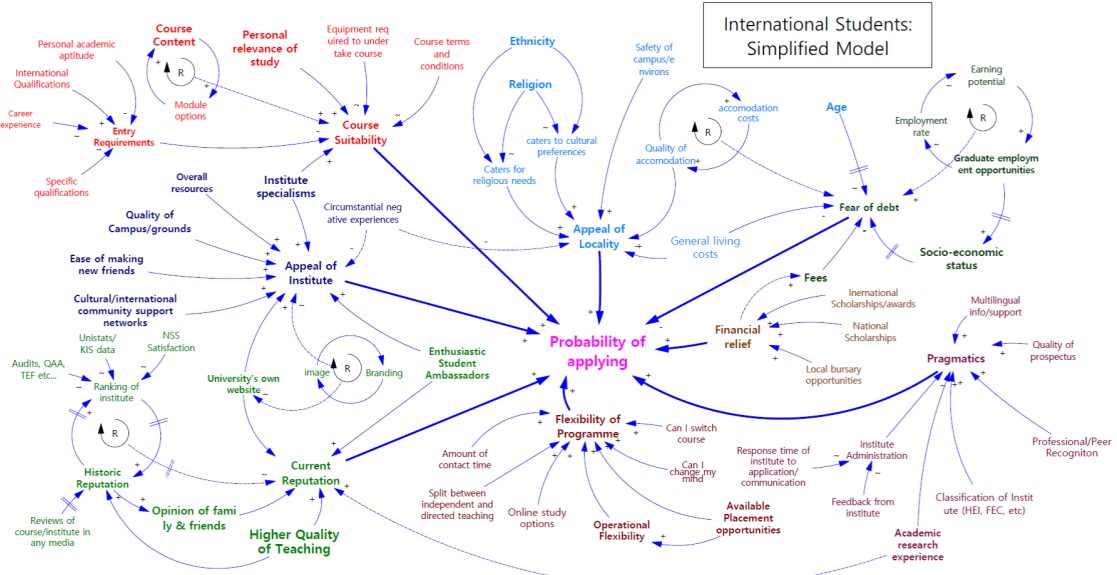

**Figure 6.** Final CLD model—international students.

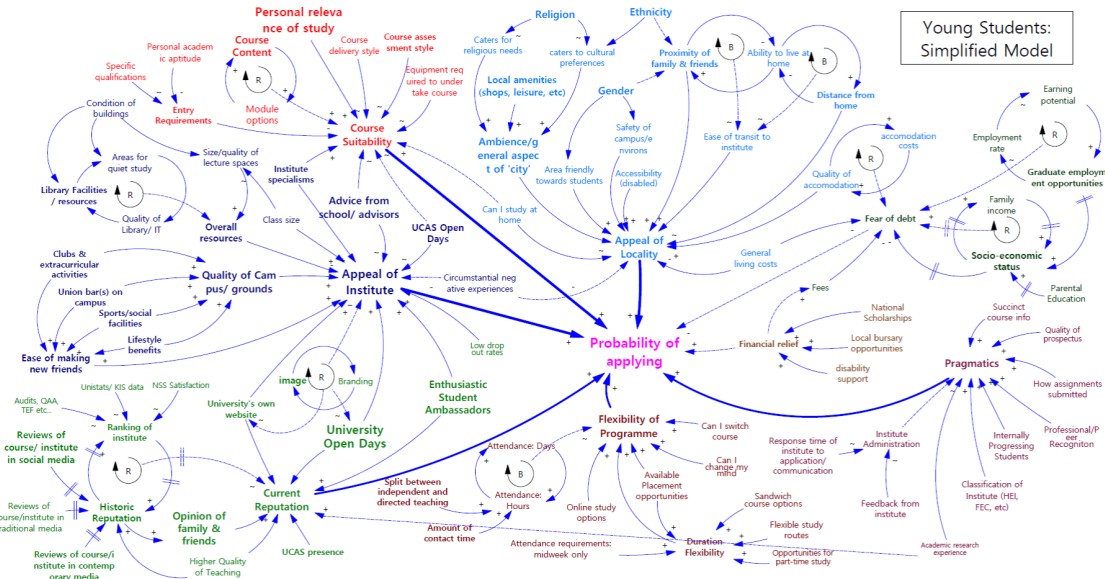

**Figure 7.** Final CLD model—young students.

*6.1. Gaps in Understanding, Uncertainties in the System*

Whilst the contributions from the interview group were extensive and valuable in expanding and developing the model, it remained the case that these were from a limited range of institutes at the lower end of the provider scale. Metropolitan universities and college-based higher education providers are only a percentage of the sector as a whole. The remainder, encompassing the largest and finest institutes in the UK, as well as specialist providers, were not engaged by the consultation, and so their perspectives can only be inferred by the initial research conducted by other studies. As this is used to inform the context of the investigation, and it was upon this that the first iteration of the model was developed, it is recognised that there is some extensive bleeding through of information. A direct confirmation of the perspective from the top of the provision rankings would have greatly enhanced this material.

It was a cognitive choice of the study to concentrate on English institutes, but it is recognised that the regional factors that influence the model may well be magnified when looking at the home nations, most especially Scotland. The Scottish situation is distinct, as there are hugely different fee regulations for native Scots, and the level of support there is an enormous overriding factor in student choice; for Scottish natives, university is charged at a nominal, contributory level—up to £1820 per annum in 2017—which can, in many cases, be wholly subsidised by bursaries, whilst non-natives will pay broadly similar fees to those of English institutes [37]. The impact of this on the decision process, in hand with the distinct regional distances involved, is certain to be a factor driving Scottish students to remain in their home nation whenever possible. Whilst fees are less of a factor, it is expected that there would also be distinguishing variables related to geography for Welsh and Northern Irish applicants.

Additionally, the international student question was not prioritised in early development, although interesting material came to light in the midst of the design process; this is consequently a factor considered in the final and the third iteration of the model. For the most part, interviewees did not expand on the international factor greatly, though this is no doubt largely because the subject was not extensively pursued in the questioning.

Another element that proves difficult to quantify is the impact of intangible factors. Several of the interviewees discussed anecdotal cases where decisions turned on events entirely outside of an institute's control, but it remains contentious as to whether these realistically are a variable to consider for all applicants, or whether the model could be written off wholesale for a small random group of students for whom such incidences would take precedence over any other factor. Naturally, this rails

against the analytical expectations of regulators, and it seems difficult to credit elements—that are essentially luck-based—as being significant enough to drive students from the extremes of choice; would an unpleasant interview experience, possibly unrelated to the institute itself, put an applicant off their first choice of provider? However, it appears this is a widely accepted scenario, and one with some case evidence and much anecdotal material to support it. A decision on how to record this had to be made, and ultimately it was decided to recognise it as a variable, but not to give it disproportionate importance. In many cases, such experiences simply will not occur, or will be so insignificant as to have a negligible effect on the model as a whole.

*6.2. Misdirected Aims: The Fallacy of a Data-Led Decision Process*

It has become increasingly clear that the government policy towards operating Higher Education providers along the same lines as statutory providers is a mixed blessing. The initial development of Unistats and the KIS dataset was part of a firm belief that students needed to be objective 'maximisers' in the process of selecting an institute, and so sought to provide an extensive range of comparable data to meet this remit. The public face of this policy approach remains in place, with only limited changes to the KIS expected; after consultation, the KIS will officially be renamed Unistats from autumn 2017, the biggest effective change being that institutes will be responsible for hosting (and better presenting) academic data on their own websites, but the volume of recorded material is unlikely to actually change.

The Teaching Excellence Framework (TEF) takes over as the benchmark of performance for institutes—reflecting only six key measures from the KIS, NSS, and institutional material—determining the fee-charging cap and consequent delivery criteria of institutes. However, the TEF will not easily be available to the public, but the results will rather be filtered through the national rankings, each potentially with its own agenda. The Times Higher Education efforts to analyse institute performance based on the proposed measures showed a huge slip in relative performance for research institutes, whilst in the wider TEF divisions as a whole, most institutes come out in what will ultimately be viewed as the bottom tier of performers. This may meet a notional benchmark for now, but how long until a three-tier system is deemed unfit and in need of more detail?

However, the vast majority of research supports the argument that data, performance information, and statistics fall very low on the range of factors that drive student choice. Therefore, why do the regulations remain married to a largely unproven thesis? One possible reason is that the alternatives are too difficult to quantify, are naturally challenging to measure, and, hence, are difficult to benchmark. Ephemeral variables, such as the social, cultural, or environmental appeal of any given institute, are well beyond the capabilities of traditional performance management models to measure. There has to be a recognition that, as important as these elements may be, they remain an art to quantify rather than a science. However, with a demand to provide some form of objective data as a counter to provider marketing, HEFCE finds itself committed to the ongoing use of qualitative data of little realistic value to consumers. Another reason simply seems to be the entrenched culture of regulation of the past thirty or more years, ironically driven by the ability of information technology to revolutionise data collection and collation.

However, these data do ably serve the other demands of the regulator—to delineate institutional subscription to standards required by the government for academic provision. Without such data and the wider regulations that they report to, institutes would be free to report any material they saw fit, with any interpretation. In this respect, performance data seek to support the quality auditing of education, but the error, if there is one, is in the belief that students would certainly also seek to use these data as part of their selection procedure. The production of the conceptual causal model has resulted in the consolidation of a number of research resources with field experience from a variety of expert elicitations. This has led to a range of findings. These include that students use a wide variety of sources of information and operate subject to a number of peer influences and instinctive or contextual preferences, which may not be openly expressed. Conversely, the corporate, standardised data pushed

for by regulators is largely overlooked, or at least considered a low priority by most of this group. Students seem to place the most priority on a handful of key factors:

- Personal interest in field of study;
- Entry tariff;
- Appeal of city/region of institute—particularly in how it may relate to present living requirements;
- Appeal of institute itself—particularly with relation to hands-on experience of site visits;
- Reputation and any perception of quality;
- Resources and facilities—especially for more specialist provision (sport, sciences, arts, etc.).

However, around these, other variables swirl, with varying influence based on students' specific experiences. Against these, supposedly important variables that, on closer examination, appear to have little influence include:

- Fees;
- Academic research reputations;
- Retention and outcome rates;
- Availability of in-depth course/institute structures and practices in advance;
- Ranking of institutes based on performance data.

Naturally, again, exceptions always occur, and certain contexts can be attributed to the clientele of individual institutes; academic research is more likely a factor for students applying to Russel Group universities, but they will (for rather different reasons, no doubt) have a disdain for comparing fees similar to that of the student of a post-1992 institute. That students' priorities can be predicted as much by demographic properties as by raw academic interest should not come as a surprise, yet the regulatory information has largely sought to operate in a bubble untouched by such contextualising. There are nevertheless clear research influences with a strong demographic element that point to such contexts as a huge part of the overall picture; UCAS data for 2015 makes a strong case for Black and other minority students applying to lower-tariffed institutes than their grades alone would have predicted [38], whilst elsewhere, it is established that their grade predictions are typically less accurate than for white students; thus, a cyclical reinforcement can occur, where lower attainment is, in part, expected, and lower aspiration follows [39]. Like for like, a BME (Black and Minority Ethnic) student will aim lower than a similar white student; applying raw data as a solution to the application process will take no account of this, and will thus fail to address such variables.

### 6.3. What Information Should Institutes Focus On?

As shown in the previous section, a number of priorities exist. These are not all within an institute's power to control, but they can be recognised as the most important in providing information on if a student decision is to be supported. Regulations indeed support the provision of course data, and it is certainly in institutes' interests to be accurate and succinct on this. Current requirements for information presented by the UK Quality Assurance Agency for Higher Education (2017) are a sensible and comprehensive opening standpoint [6], but may be prone to a surfeit of detail compared to what experts considered useful and students expressed actual use of.

Issues of institute profiles, reputation, and campus/environment experience are generally viewed as a function of marketing, but there appears to be an academic disconnect here—not all recognise the value of incorporating such material into the course itself. Institutes need to understand what it is that typical students come to their institute for, and though this may relate to fields of specialism, it should not be assessed at the granular scale of individual courses. Such a specific detail aside, it is of more value to understand the groups within the student body, and to see how these relate to patterns of behaviour. An institute largely recruiting from high-tariff students outside of its region will be wholly different from another attracting low-tariff students, even to similar courses, in the same location.

### 7. Conclusions

The core aim of this study was to design an iterative model of variables involved in the student application decision process, and, by extension, to use this as a means to investigate the use of data in performance reporting and recruitment for Higher Education Providers nationally. By addressing this through a series of Soft Systems Methodologies, the intent was to show how IT-derived business theories could result in better-implemented knowledge in the sector and allow for a fairer measurement of performance at a national level, resulting in a better environment for both institutes and students to operate within. As a factor, institutional reputation could be influenced by several other elements from within the HE system, its environment, and the wider environment. In this piece of research, this factor is not further expanded upon.

Factors such as student age, race, disability status, gender, and religion are overarching influences on a huge number of other variables. The subtle interplay of these and the interrelations of variables offer near infinite combinations, but the modelling of the connectivity is a recognition of the complexity of the model, which raw data and performance information alone have not addressed. This is what makes such modelling so valuable—its ability to visualise patterns of behaviour that are otherwise occluded by opaque statistics, however well defined they may be.

It is hoped that these aims have all been met to some degree, but it is recognised that no one solution can be comprehensive or perfect, and, in particular, there are a number of areas open to further refinement. Moreover, additional research work to capture students' perspectives is encouraged. Other respondents, such as academic and non-academic staff, regulators, parents, etc., could also add their perspectives to the model. The proposed model encapsulates the observed students' behavioural factors from the providers' perspective; nonetheless, students' and other stakeholders' inputs can also be considered as a validation medium for the proposed model.

The modelling and visualisation approach used in this study is based on causal loop diagramming solely to represent the causalities and interconnections between the identified variables and to simplify the complexities [40] that the HE system entails. Nevertheless, for future works, the causal loop diagrams can be converted into stock flow diagrams for quantitative simulation. Moreover, the study proposes that a better design for data feedback and resultant policy development is needed.

**Author Contributions:** Conceptualisation, D.H.; methodology, A.H.-F.; investigation, D.H.; writing—original draft preparation, D.H.; writing—review and editing, D.H., S.S., and A.H.-F.; visualisation, D.H.; supervision, A.H.-F.; project administration, D.H. All authors have read and agreed to the published version of the manuscript.

**Funding:** This research received no external funding.

**Conflicts of Interest:** The authors declare no conflict of interest.

### Abbreviations

The following abbreviations are used in this manuscript:

| | |
|---|---|
| CATWOE | Customers, Actors, Transformation, Weltanschauung, Owner, and Environment |
| CBHE | College-Based Higher Education |
| DSR | Design Science Research |
| FE | Further Education |
| HE | Higher Education |
| HEFCE | Higher Education Funding Council for England |
| HEPI | Higher Education Policy Institute |
| KIS | Key Information Set |
| NSS | National Student Survey |
| QAA | Quality Assurance Agency |
| TEF | Teaching Excellence Framework |
| UCAS | The Universities and Colleges Admissions Service |



**Appendix A. Interview Questions**

(1)     Firstly, a little about yourself. What is your job title, which department do you sit within, and how would you sum up your role in relation to student recruitment (bearing in mind the relationship may be tangential)?

☐ I hold a senior Executive role (Director, Provost, Chief Executive, Director of Higher Education, etc.).

☐ I am the Chief Registrar, Head of Admissions, or a similar senior role.

☐ I am an Information Manager, Analyst, or other data lead for Higher Education.

☐ I hold a senior Marketing or Market Research role.

☐ I am an admissions officer or student liaison representative.

☐ Other; please specify:

(2)     Thinking about your own Higher Education (HE) provision and how it appeals to an audience, what do you think are the priorities for your students in selecting a course with your institute?

(3)     How do you think this may vary from national trends?

(4)     What is your opinion on the quality of data available to prospective students in the following public arenas?

   (i)     Your institute's web presence.
   (ii)    UNISTATS and the Key Information Set (KIS) dataset in general.
   (iii)   HE Provider rankings and comparator sites (e.g., the Times HE Rankings, Which? University).

(5)     Do you feel adequate context is placed within publicly available performance data? How might you change it?

(6)     Looking at the derived list of students' priorities—used to drive the CLD produced—you will have an indication of how students actually prioritise variables in their choice of institute. Although academic factors rank top, the weighting indicates that other personal and demographic factors are more important than often supposed. Therefore:

   (i)     Are you surprised by any of the variables within the list; their position, or the fact they are factors at all?
   (ii)    What variables would you expect to see on the list that are missing?
   (iii)   Which variables do you feel your institute overlooks that appear important to student choice?
   (iv)    Which variables, that do not appear as important to student choice, do you feel your institute does, or should, focus on?

(7)     Thinking about the CLD and the associated data, how comprehensible do you find this?

☐ Entirely.

☐ Comprehensible, but I need some clarifications.

☐ Not sure, this is a new concept to me.

☐ No, it is not clear at all to me.

(8)     Looking specifically at the CLD:

   (i)     Are the any connections that you found particularly surprising?
   (ii)    Are there any omissions in terms of links or priorities?

(9)     Regulations and standardised rankings, such as the KIS dataset, the National Student Survey (NSS), and the Teaching Excellence Framework (TEF), highlight performance data. Do you feel the publication of more data would be helpful to prospective students? How would you feel if at

some point in the future, the TEF introduced a larger number of data points into its measurement?

(10) From an institutional position, would it be more appropriate to present derived information that allowed the institute to present a more balanced perspective of itself?

(11) Would you like to see such information more generally used for comparison of providers? Why?

(12) If the following hypothetical scenarios were to take place at your institute, how do you feel they would impact your student applications?

   (i) The institute dropped entry criteria by the equivalent of one 'B' grade in A-Levels (40 pts); explain your answer in each case.

      Boom,    Increase,    No effect,    Decrease,    Crash

   (ii) The institute dropped fees by £1500.

      Boom,    Increase,    No effect,    Decrease,    Crash

   (iii) The institute fell into the bottom quartile in a major national institute ranking.

      Boom,    Increase,    No effect,    Decrease,    Crash

(13) Lastly, what pressures do you envisage on your institute in the next five years regarding recruitment?

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
