# Peer review of "Factors Affecting University Choice Behaviour in the UK Higher Education"

_education, doi:10.3390/educsci10080199_

Round 1
Reviewer 1 Report
I received a no blind version of the article. I can not go on with my review.
Author Response
No review has been provided for reviewer 1.
Reviewer 2 Report
in these days, education has become more important due to various modern approaches such as Industry 4.0, Society 4.0, Leadership 4.0, or Education 4.0. Individual education systems need to be developed according to present global requirements.
In the submitted paper, there are several errors to remove:
The paper does not match the template of the journal.
The format of references does not meet the requirements of the journal. If it is possible, I recommend adding the DOI number to used sources.
At the same time, the authors mix two quotation styles.
The authors write, that they used an interview as a research method, but they do not mention a description of interviewers. At the same time, there is not clear if realized five interviews were focused on three specific areas as they prepared individual models (students, mature students, and international students). Based on one group is not possible to develop three different models. In chapter 4.1 (or anywhere else) is no description of interviewers.
Chapter 5 described specific scenarios of education. According to the explanation, there should be very suitable possible quantification, especially, if authors write "..most interviewees.. " (L342). In the context of chapter 4.2, it is difficult to say "most interviewees" if there were realized 5 interviews in three rounds.
Also, I recommend extending the theoretical background with newer sources.
Author Response
We would like to thank you for taking the time in reading our manuscript. We also believe that the points in your feedback are very helpful and will certainly benefit the quality of our work; therefore, there is no rebuttal on the points raised.
Review: In these days, education has become more important due to various modern approaches such as Industry 4.0, Society 4.0, Leadership 4.0, or Education 4.0. Individual education systems need to be developed according to present global requirements.
In the submitted paper, there are several errors to remove:
The paper does not match the template of the journal.
Response to point 1:
In response to point 1, we would like to highlight that we have used the MDPI Overleaf Latex template (https://www.mdpi.com/authors/latex); due to a technical glitch our template was not loading the Journal’s footer in the original submission. Please note that the issue is now rectified by adding the relevant command in the template: (\documentclass[education,article,submit,moreauthors,pdftex]{Definitions/mdpi}) .
We have also received a confirmation from the Assistant Editor: “… Please make sure that you mention that you are using the correct template in your revision report…”.
The guidelines provided with the template also highlights that “The class option "submit" will be changed to "accept" by the Editorial Office when the paper is accepted. This will only make changes to the frontpage (e.g., the logo of the journal will get visible), the headings, and the copyright information. Also, line numbering will be removed. Journal info and pagination for accepted papers will also be assigned by the Editorial Office.”
The format of references does not meet the requirements of the journal. If it is possible, I recommend adding the DOI number to used sources.
At the same time, the authors mix two quotation styles.
Response to point 2:
Thank you for this point. We have amended the citations and the full references to make the referencing style consistent and in-line with the publisher’s requirements.
The authors write, that they used an interview as a research method, but they do not mention a description of interviewers. At the same time, there is not clear if realized five interviews were focused on three specific areas as they prepared individual models (students, mature students, and international students). Based on one group is not possible to develop three different models. In chapter 4.1 (or anywhere else) is no description of interviewers.
Response to point 3:
Thank you for your comment. We have now created a new section (Section 4.5) to include the interviewees’ profiles. The following text has been therefore included within the manuscript:
“4.5 Interviewee Profiles
The five interviewees all worked within education, in a variety of roles, and represented a range of institutes:
- A Course Leader and Teaching Fellow for a regional metropolitan university in northern England
- A Principal Lecturer, coordinating the recruitment activities of their faculty at a regional metropolitan university in northern England
- The Head of Marketing and Recruitment for FE and HE provision at a northern regional college in England
- The Director of Higher Education Participation and Skills for a metropolitan college-based HE provider in the northeast England; recently of a role with another CBHE provider in one of the country’s largest cities
- The Market Research Manager for a regional metropolitan university in northern England
This permitted a range of perspectives to be drawn on the content of the CLD and the working context it attempted to represent. It is notable that the group, being of a small size, does not contain a full diversity of institute types, comprising post-1992 institutes and college based higher education providers. There are no older ‘Plate Glass’ or ‘Redbrick’ providers’ in the set, and certainly no identifiably premium institutes. Red Brick universities are institutes from approximately the 1950’s to 1990’s and from the late 19th to mid-20th century respectively. One concern for the study was the difficulty in engaging institutes themselves in a relatively short timeframe. 63 staff across 43 institutes were approached with only 5% engagement rate.”
Moreover, with respect to the divergence of the single perspective model for all students into three subtype groups, the following clarification was also added in line 509:
“As highlighted earlier, it became apparent that a single perspective to the factors influencing students' choice is unable to embrace all specific requirements of the subtype. Therefore, three additional subtype models were developed to entail the minor differences that cater for specific types of students.”
Please also note the necessity to develop three versions had been described in lines 500 to 506.
Chapter 5 described specific scenarios of education. According to the explanation, there should be very suitable possible quantification, especially, if authors write "..most interviewees.. " (L342). In the context of chapter 4.2, it is difficult to say "most interviewees" if there were realized 5 interviews in three rounds.
Response to point 4:
Thank you for this point. We have now made the sentence more specific and clearer by using a value rather than the soft term ‘most. Change are made to lines 353 and 458.
Also, I recommend extending the theoretical background with newer sources.
Response to point 5:
Thank you for your feedback. We have now attempted to substantiate the arguments with further current sources. These new additions are scattered throughout the manuscript, especially where new additions are made. We now have 8 additional citations that are from 2019 and 2020. The hyperlinks in all references are also checked to ensure that there is no orphan link left with in the manuscript.
All the changes we have mentioned above are reflected in the revised version in ‘Blue’ font. In light of other reviewers’ feedback, we have also made further changes that can be accessed in the attached file.

Reviewer 3 Report
A fascinating paper that touches on an area of interest to the HEI sector. The choice and application of DSR is of merit and produces valuable insights via the model that affirms the everyday wisdom of practitioners and the flaws with reductionist views of student choice. The model versions for mature, international and young students was a great addition.
Areas that require further clarification:
- Section 4.1: please explain how 'narrative extraction' was performed
- Section 4.2: more information about the 'five key interviews' is needed e.g. roles as per Q1 of Appendix A interview questions, how sampled, were they all at the same university.
Author Response
We would like to thank you for taking the time in reading our manuscript, the encouraging comments. We also believe that the points in your feedback are very helpful and will certainly benefit the quality of our work; therefore, there is no rebuttal on the points raised.
A fascinating paper that touches on an area of interest to the HEI sector. The choice and application of DSR is of merit and produces valuable insights via the model that affirms the everyday wisdom of practitioners and the flaws with reductionist views of student choice. The model versions for mature, international and young students was a great addition.
Areas that require further clarification:
Section 4.1: please explain how 'narrative extraction' was performed
Response to point 1:
Thank you very much for your feedback. We have used CATWOE analysis with a view to use it as a framework to develop the identify themes and to categorise the identified factors. Therefore, we have added the following section in the revised version of the manuscript to make this clearer:
“4.2. The Consultation Process
The narrative extraction approach taken was to develop an initial model based on the CATWOE (Customers, Actors, Transformation, Weltanschauung, Owner & Environment) analysis of primary research. CATWOE is a fundamental analysis technique within the Soft Systems Methodology approach. As a problem evaluation tool, it seeks to represent all of the factors influencing a scenario [28]. This built a test diagram within Vensim, seeking to make an initial attempt to encompass all of the identified variables, and to project their initial structural form and relative importance to the design, and consequently to the decision process.
This proposed design, along with tabular representations of the factors – attempting to reflect the relative importance of the positions of variables within major factorial groups – was then provided to the consultative members in advance of the interview process. To provide them with an opportunity to examine the design. Whilst provided in a Vensim model format as a matter of course, a clone of the diagram was also provided in an accessible format (i.e. MS PowerPoint, via image shots with supporting dialogue) for those without access to or familiarity with the software.
Consultation on the model was conducted as part of a three-stage interview process, with the second part of the dialogue concentrating on the model itself. This was preceded by discussion of the interviewees’ conception of the operational market, and followed with discussion of information use in the academic recruitment market, and some investigation of how they feel specific institutional decisions may impact student interest.
Fundamentally, the consultation over the proposed framework with numerous stakeholders within the HE sector will fulfil the ‘demonstration’ and ‘evaluation’ stages of DSR methodology. Once initial consultation is completed the design will pass through a second iteration, returned to the group for further consideration; now informed by the discourse at the first stage, as well as incorporating their recommendations and thoughts on the design. Additional questioning will be developed at this stage to address the findings of the primary interview and the evolving model. This will result in the final framework.”
Moreover, a colour-coding legend is added after line 261 in the new version of the manuscript to specify where the variables are derived from literature.
Section 4.2: more information about the 'five key interviews' is needed e.g. roles as per Q1 of Appendix A interview questions, how sampled, were they all at the same university
Response to point 2:
Thank you again for your comment. We have now created a new section (Section 4.5) to include the interviewees’ profiles. The following text has been therefore included within the manuscript:
“4.5 Interviewee Profiles
The five interviewees all worked within education, in a variety of roles, and represented a range of institutes:
- A Course Leader and Teaching Fellow for a regional metropolitan university in northern England
- A Principal Lecturer, coordinating the recruitment activities of their faculty at a regional metropolitan university in northern England
- The Head of Marketing and Recruitment for FE and HE provision at a northern regional college in England
- The Director of Higher Education Participation and Skills for a metropolitan college-based HE provider in the northeast England; recently of a role with another CBHE provider in one of the country’s largest cities
- The Market Research Manager for a regional metropolitan university in northern England
This permitted a range of perspectives to be drawn on the content of the CLD and the working context it attempted to represent. It is notable that the group, being of a small size, does not contain a full diversity of institute types, comprising post-1992 institutes and college based higher education providers. There are no older ‘Plate Glass’ or ‘Redbrick’ providers’ in the set, and certainly no identifiably premium institutes. Red Brick universities are institutes from approximately the 1950’s to 1990’s and from the late 19th to mid-20th century respectively. One concern for the study was the difficulty in engaging institutes themselves in a relatively short timeframe. 63 staff across 43 institutes were approached with only 5% engagement rate.”
All the changes we have mentioned above are reflected in the revised version in ‘Blue’ font. In light of other reviewers’ feedback, we have also made further changes that can be viewed in the attached file.

Reviewer 4 Report
The manuscript entitled “Factors Affecting University Choice Behaviour in the UK Higher Education “ is an interesting approach to evaluate choice behaviour of students in the UK higher education system. However, there is no initial question at the end of the introduction unless in the beginning of conclusion section. Also, there was no apparent justification of what was the motivation problem for this evaluation. On the other hand, there is no other indication about further options either students or institutions can follow. Thus, the manuscript has some flaws that need to be solved before it can be accepted.
Author Response
We would like to thank you for taking the time in reading our manuscript. We also believe that the points in your feedback are very helpful and will certainly benefit the quality of our work; therefore, there is no rebuttal on the points raised.
The manuscript entitled “Factors Affecting University Choice Behaviour in the UK Higher Education “ is an interesting approach to evaluate choice behaviour of students in the UK higher education system. However, there is no initial question at the end of the introduction unless in the beginning of conclusion section.
Response to point 1:
Thank you again for your feedback. We have now provided new additions to the ‘Introduction’ and ‘Background & the Problem Domain’ sections to better articulate the gap, by also including the original research question. The following sections are now included in the two above-mentioned sections respectively:
“This piece of research attempts to identify and structure the factors that are affecting student choice in the UK higher education system. It also offers methodological novelty by adopting the Design Science Research (DSR) methodology to visualise a causal model, embracing all the identified factors. This paper is organised as follows: Section 2 provides the research background and elaborates on the problem domain. Section 3 similarly provides a review of key literature with a view to identify the variables for the initial model. Sections 4 and 5 outline the DSR methodology, and where required, reflect on some of the key overall responses as part of the interview data collection method. Results including the final models, and a discussion on the models are provided in Section 6. The paper is book-ended by a Conclusion section in Section 7. “
“Despite that the regulations and established practices in academia have focused on a data rich model of performance information to evidence operational capability and to support recruitment, yet this data-led approach is widely argued to be ineffective in addressing students' choice behaviour [8, 9]. Therefore, this piece of research seeks to answer the following research question; what factors interplay with the student decision making process? Students are in this treatment seen as the heart of the decision making cypher and so it is vital to survey the problem of choice from their standpoint, ultimately any actions by external bodies can only influence their action, never dictate it. By this logic, for the purposes of this study, the prospective student stands as the primary ‘Customer’ within the system. But this is not to say that institutes cannot be viewed as the beneficiaries of the model as a whole in their own way too.”
Also, section 2.2. has three additional sentences that improve the flow of argument within the section.
Also, there was no apparent justification of what was the motivation problem for this evaluation. On the other hand, there is no other indication about further options either students or institutions can follow. Thus, the manuscript has some flaws that need to be solved before it can be accepted.
Response to point 1:
Thank you for this point. We have strengthened our discussion in relation to the findings in order to better articulate the benefits from causal models such as ours. These additions elucidate institutional priorities and the policies that can shape the future of data-led strategies.
Moreover, the following section was added to further clarify our argument on the misconceptions on data-led strategies and the need for richer understanding of the sector:
“6.2 Misdirected Aims; the Fallacy of a Data-led Decision Process
What has become increasingly clear that the government policy towards operating Higher Education providers along the same lines as statutory providers is a mixed blessing. The initial development of Unistats and the KIS dataset was part of a firm belief that students needed to be objective ‘maximisers’ in the process of selecting an institute, and so sought to provide an extensive range of comparable data to meet this remit. The public face of this policy approach remains in place, with only limited changes to the KIS expected; after consultation the KIS will officially be renamed Unistats from autumn 2017, the biggest effective change being that institutes will be responsible for hosting (and better presenting) academic data on their own websites, but the volume of recorded material is unlikely to actually change.
The TEF takes over as the benchmark of performance for institutes – reflecting only 6 key measures from the KIS, NSS and institutional material – determining the fee charging cap, and consequent delivery criteria of institutes, however the TEF will not easily be available to the public, but the results will rather be filtered through the national rankings, each potentially with its own agenda. The Times Higher Education efforts to analyse institute performance based on the proposed measures showed a huge slip in relative performance for Research Institutes, whilst in the wider TEF divisions as a whole most institutes come out in what will ultimately be viewed as the bottom tier of performers. This may meet a notional benchmark for now, but how long until a three-tier system is deemed unfit and in need of more detail?
Yet the vast majority of research supports the argument that data, performance information, statistics fall very low on the range of factors that drive student choice. Why do the regulations remain married to a largely unproven thesis therefore? One possible reason is that the alternatives are too difficult to quantify, and naturally challenging to measure, hence difficult to benchmark. Ephemeral variables such as the social, cultural or environmental appeal of any given institute are well beyond the capabilities of traditional performance management models to measure. There has to be a recognition that as important as these elements maybe they remain an art to quantify rather than a science; but with a demand to provide some form of objective data as a counter to provider marketing, HEFCE finds itself committed to the ongoing use of qualitative data of little realistic value to consumers. Another simply seems to be the entrenched culture of regulation of the past thirty or more years, ironically driven by the ability of information technology to revolutionise data collection and collation.
However, this data does, ably, serve the other demands of the regulator, to delineate institutional subscription to standards required by the government for academic provision. Without such data, and the wider regulations it reports to, institutes would be free to report any material they saw fit, with any interpretation. In this respect performance data seeks to support the quality auditing of education; but the error, if there be one, is in the belief that students would certainly also seek to use this data as part of their selection procedure. The production of the conceptual causal model has resulted in the consolidation of a number of research resources with field experience from a variety of expert elicitations. This has led a range of findings. These include that students use a wide variety of sources of information, and operate subject to a number of peer influences and instinctive or contextual preferences, that may not be openly expressed. Conversely the corporate, standardised, data pushed for by regulators is largely overlooked, or at least considered a low priority, for most of this group. Students seem to place most priority on a handful of key factors:
- Personal interest in field of study
- Entry Tariff
- Appeal of city/region of institute – particularly in how it may relate to present living requirements
- Appeal of institute itself – particularly with relation to hands on experience of site visits
- Reputation and any perception of quality
- Resources and facilities – especially for more specialist provision (sport, sciences, arts, etc.)
But around these, other variables swirl, with varying influence based on a students’ specific experiences. Against these, supposedly important variables that on closer examination appear to have little influence, include:
- Fees
- Academic Research reputations
- Retention and Outcome rates
- Availability of in-depth course/institute structures & practices in advance
- Ranking of institutes based on performance data
Naturally again, exceptions always occur, and certain contexts can be attributed to the clientele of individual institutes; academic research is more likely a factor for students applying to Russel Group universities, but they will (for rather different reasons no doubt) have a similar disdain for comparing fees as does the student of a post-1992 institute.
That students’ priorities can be predicted as much by demographic properties as by raw academic interest should not come as a surprise, yet the regulatory information has sought largely to operate in a bubble untouched by such contextualising. There are nevertheless clearly research influences with a strong demographic element that point to such context as a huge part of the overall picture; UCAS data for 2015 makes a strong case for Black and other minority students applying to lower tariffed institutes than their grades alone would have predicted [38], whilst elsewhere it is established that their grade predictions are typically less accurate than for white students; thus, a cyclical reinforcement can occur where lower attainment is in part expected and lower aspiration follows [39]. Like for like, a BME (Black and Minority Ethnic) student will aim lower than a similar white student; applying raw data as a solution to the application process will take no account of and thus fail to address such variables.”
Moreover, we have strengthened the conclusion section to summarise the key factors and also an additional future work:
“Factors such as student age, race, disability status, gender and religion are overarching influences on a huge number of other variables. The subtle interplay of these, and the interrelations of variables offer near infinite combinations, but the modelling of the connectivity is a recognition of the complexity of the model, which raw data and performance information alone has not addressed. This is what makes such modelling so valuable, its’ ability to visualise patterns of behaviour that are otherwise occluded by opaque statistics, however well-defined they may be.”
“Moreover, the study proposes that a better design for data feedback and resultant policy development is needed.”
All the changes we have mentioned above are reflected in the revised version in ‘Blue’ font. In light of other reviewers’ feedback, we have also made the following changes and/or additions:

Round 2
Reviewer 2 Report
Authors correct defined errors, which were mentioned in the first review round. At this moment, the paper reaches a standard level of sciešntific papers and could be published.
Reviewer 4 Report
The present manuscript is much better. The improvement was considerable and I do think it is ready to be accepted.